# The Critical Role of NLRP6 Inflammasome in *Streptococcus pneumoniae* Infection In Vitro and In Vivo

**DOI:** 10.3390/ijms22083876

**Published:** 2021-04-08

**Authors:** Dongyi Xu, Xingping Wu, Lianci Peng, Tingting Chen, Qingyuan Huang, Yu Wang, Chao Ye, Yuanyi Peng, Dongliang Hu, Rendong Fang

**Affiliations:** 1Joint International Research Laboratory of Animal Health and Animal Food Safety, College of Veterinary Medicine, Southwest University, Chongqing 400715, China; xudongyi29@gmail.com (D.X.); yangyusisier@163.com (X.W.); penglianci@swu.edu.cn (L.P.); cttctt@outlook.com (T.C.); huangqingyuan2020@163.com (Q.H.); wyseven67@163.com (Y.W.); yechao123@swu.edu.cn (C.Y.); pyy2002@sina.com (Y.P.); hudl@vmas.kitasato-u.ac.jp (D.H.); 2Department of Zoonoses, School of Veterinary Medicine, Kitasato University, Towada 034-8628, Japan; 3Immunology Research Center, Medical Research Institute, Southwest University, Chongqing 402460, China

**Keywords:** *Streptococcus pneumoniae*, NLRP6, inflammasome, IL-1β, inflammatory response

## Abstract

*Streptococcus pneumoniae* (*S. pneumoniae*) causes severe pulmonary diseases, leading to high morbidity and mortality. It has been reported that inflammasomes such as NLR family pyrin domain containing 3 (NLRP3) and absent in melanoma 2 (AIM2) play an important role in the host defense against *S. pneumoniae* infection. However, the role of NLRP6 in vivo and in vitro against *S. pneumoniae* remains unclear. Therefore, we investigated the role of NLRP6 in regulating the *S. pneumoniae*-induced inflammatory signaling pathway in vitro and the role of NLRP6 in the host defense against *S. pneumoniae* in vivo by using NLRP6^−/−^ mice. The results showed that the NLRP6 inflammasome regulated the maturation and secretion of IL-1β, but it did not affect the induction of IL-1β transcription in *S. pneumoniae*-infected macrophages. Furthermore, the activation of caspase-1, caspase-11, and gasdermin D (GSDMD) as well as the oligomerization of apoptosis-associated speck-like protein (ASC) were also mediated by NLRP6 in *S. pneumoniae*-infected macrophages. However, the activation of NLRP6 reduced the expression of NF-κB and ERK signaling pathways in *S. pneumoniae*-infected macrophages. In vivo study showed that NLRP6^−/−^ mice had a higher survival rate, lower number of bacteria, and milder inflammatory response in the lung compared with wild-type (WT) mice during *S. pneumoniae* infection, indicating that NLRP6 plays a negative role in the host defense against *S. pneumoniae*. Furthermore, increased bacterial clearance in NLRP6 deficient mice was modulated by the recruitment of macrophages and neutrophils. Our study provides a new insight on *S. pneumoniae*-induced activation of NLRP6 and suggests that blocking NLRP6 could be considered as a potential therapeutic strategy to treat *S. pneumoniae* infection.

## 1. Introduction

*Streptococcus pneumoniae* (*S. pneumoniae*) is a Gram-positive extracellular bacteria causing severe infection in the respiratory tract, leading to high morbidity and mortality worldwide, especially among children. It has been reported that *S. pneumoniae* causes at least 1.2 million infant deaths every year worldwide [1]. *S. pneumoniae* is an opportunistic bacterial pathogen and causes invasive pneumococcal diseases such as community-acquired pneumonia, sepsis, meningitis, and otitis media [2]. *S. pneumoniae* often colonizes on the mucosal surface of the upper respiratory tract and the dynamic process makes this pathogen invade the lower airways [3,4]. The host in turn produces a series of immune responses including inflammatory response against *S. pneumoniae* infection [5,6]. Therefore, understanding the host response to *S. pneumoniae* is essential to prevent and treat *S. pneumoniae* infection.

The innate immunity plays an important role in the host defense against pathogens in the early stage of infection. The innate immune response is regulated by different pattern recognition receptors (PRRs) in the immune cells, such as toll-like receptors (TLRs), RIG-I-like receptors (RLRs), C-type lectin receptors (CLRs), and NOD-like receptors (NLRs) [7]. Inflammasome is a member of NLRs family and has been recognized for its critical role in innate immunity against microbial infection [8,9]. To date, multiple proteins receptors have been confirmed to assemble inflammasomes, such as the nucleotide-binding oligomerization domain (NOD), leucine-rich repeat (LRR)-containing protein (NLR) family members NLRP1, NLRP3, NLRP6, NLRP7, NLRC4, and absent in melanoma 2 (AIM2) etc. NLRP3 is one of the most extensively studied inflammasomes [7,10,11]. Caspase-1 is activated in canonical inflammasomes, while related caspase-11 is activated in the non-canonical pathway via sensing of cytoplasmic lipopolysaccharide (LPS) of Gram-negative bacteria. However, the results of canonical and non-canonical inflammasome activation are similar [12]. Caspase-1 induces the processing and release of the proinflammatory cytokine interleukin (IL) 1β and IL-18, as well as a lytic form of cell death called pyroptosis. Caspase-11 directly promotes the cleavage of gasdermin D (GSDMD), which triggers a secondary activation of canonical NLRP3 inflammasome for cytokine release [13]. It has been reported that *S. pneumoniae* induced activation of NLRP3 and AIM2 in macrophages results in the maturation and secretion of IL-1β [14,15]. Our previous studies have shown that the absence of NLRP3 or AIM2 significantly reduced the host defense against *S. pneumoniae*, resulting in higher mortality and bacterial colonization in the lungs, which indicates the protective role of NLRP3 and AIM2 in the host against *S. pneumoniae* infection [12,16,17]. However, among these inflammasomes studies, the role of NLRP6 in the host is less studied.

NLRP6 has been identified to play an important role in the intestinal homeostasis [18]. For example, NLRP6 has a positive role in the intestine against *Citrobacter rodentium* infection by regulating goblet cell mucus secretion [19]. In contrast, NLRP6 also has been found to negatively regulate host defense against *Listeria monocytogenes*, *Staphylococcus aureus*, and *Salmonella* [20,21,22]. These results demonstrate that NLRP6 plays a complicated role in the host in response to different pathogens. However, it is still unclear what the role of NLRP6 is in the host against *S. pneumoniae*. 

In this study, primary mouse macrophages were used as a cell model to investigate *S. pneumoniae*-induced activation of NLRP6. The results showed that the secretion of IL-1β was partially dependent on activation of NLRP6 inflammasome during macrophages infection with *S. pneumoniae*. Then, wild-type (WT) and NLRP6^−/−^ mice were used as intranasal infection models to investigate the role of NLRP6 in the host against pneumococcal infections. Our results showed that NLRP6^−/−^ mice had lower mortality, lower bacterial colonization, and milder inflammation in the lungs compared to WT mice, suggesting that NLRP6 plays a detrimental role in the host defense against *S. pneumoniae*. Moreover, NLRP6 negatively modulated the recruitment of neutrophils and macrophages. Our study provides more information about the role of NLRP6 in the host against pulmonary microbial infections.

## 2. Results

### 2.1. NLRP6 Inflammasome Mediates Proinflammatory Cytokines Secretion during Macrophages Infection with S. pneumoniae

To investigate the effect of the NLRP6 inflammasome on the production of proinflammatory cytokines during macrophages infection with *S. pneumoniae*, mouse primary macrophages from C57BL/6 (WT) and NLRP6^−/−^ were infected with *S. pneumoniae*. After 24 h of infection, inflammatory cytokines secretion in the supernatants were determined by ELISA. The results showed that *S. pneumoniae* induced a high level of secretion of IL-1β, IL-1α, IL-6, and IL-12p40 in WT mice macrophages, while these cytokines’ secretion was significantly reduced in NLRP6^−/−^ macrophages (Figure 1). However, there was no difference in the production of TNF-α. These results indicate that the production of IL-1β, IL-1α, IL-6, and IL-12p40 is partially dependent on the activation of NLRP6 in *S. pneumoniae*-infected macrophages.

### 2.2. NLRP6 Is Involved in the Maturation and Secretion of IL-1β But Not in the Induction of IL-1β Transcription in S. pneumoniae-Infected Macrophages

It has been known that IL-1β is synthesized as a 31 kDa precursor protein (pro-IL-1β) and then processed into a 17 kDa mature form protein for secretion. To explore the mechanism of NLRP6-mediated IL-1β secretion during macrophages infected with *S. pneumoniae*, we examined *S. pneumoniae*-induced expression of IL-1β mRNA and IL-1β protein in the supernatants and cell lysates in WT and NLRP6^−/−^ macrophages. Real-time reverse transcription-PCR (RT-PCR) results showed that the expression of IL-1β mRNA was significantly induced in *S. pneumoniae*-infected macrophages of WT and NLRP6^−/−^ mice at 9 h post infection (Figure 2A), indicating that NLRP6 did not affect *S. pneumoniae*-induced IL-1β transcription. However, the protein level of IL-1β in the supernatants was significantly lower in *S. pneumoniae*-infected NLRP6^−/−^ macrophages compared with WT macrophages, but IL-1β production was significantly higher in cell lysates (Figure 2B,C). In contrast to IL-1β, *S. pneumoniae*-induced TNF-α production was not affected in NLRP6^−/−^ macrophages (Figure 2E–G). Notably, the total level of IL-1β production in the supernatants and cell lysates was almost the same in *S. pneumoniae*-infected WT and NLRP6^−/−^ macrophages (Figure 2D), indicating that the protein production of IL-1β was consistent with the pattern of mRNA expression. Thus, these data indicated that NLRP6 regulates the maturation and secretion of IL-1β but does not regulate the induction of IL-1β transcription in *S. pneumoniae*-infected macrophages.

### 2.3. NLRP6 Inflammasome Mediates Activation of Caspase-1 and Caspase-11 in S. pneumoniae-Infected Macrophages

Caspase-1 is known as an IL-1β-converting enzyme that can directly cleave pro-IL-1β into the biologically active form IL-1β and the activation of caspase-1 is induced by the formation of large intracellular associated speck-like protein (ASC) aggregates called ASC specks. Caspase-11 is also critical for the activation of caspase-1 and IL-1β secretion [21]. Furthermore, it has been shown that GSDMD is essential for caspase-11-dependent IL-1β maturation [23]. Therefore, to assess whether the activation of caspase-1, ASC, caspase-11, and GSDMD is involved in activation of NLRP6 in *S. pneumoniae*-infected macrophages, macrophages from wild-type (WT) and mutant mice were infected with *S. pneumoniae*. Western blot analysis showed that activation of caspase-1 was significantly reduced in *S. pneumoniae*-infected NLRP6^−/−^ macrophages but the activity of caspase-1 p45 in the cell lysates was not affected (Figure 3A,C). Similarly, the mature form of IL-1β was detected in culture supernatants in *S. pneumoniae*-infected WT macrophages, while it is hardly detected in the NLRP6^−/−^ macrophages (Figure 3A,B), which is consistent with the ELISA results as shown in Figure 1A and Figure 2B. Moreover, the aggregation of ASC in *S. pneumoniae*-infected WT and NLRP6^−/−^ macrophages was tested by immunoblotting. The results showed that oligomerization of ASC was significantly induced by *S. pneumoniae* in WT macrophages, but it was markedly reduced in NLRP6^−/−^ macrophages (Figure 3D), indicating that NLRP6 mediates the activation of caspase-1 and formation of ASC specks.

Next, to further test whether activation of NLRP6 inflammasome regulates the activation of caspase-11 and GSDMD, different formations of caspase-11 and GSDMD were detected by western blot. In this study, we also tested the activation of caspase-11 in *S. pneumoniae*-infected NLRP3^−/−^ and ASC^−/−^ macrophages. The analysis showed that the induction of a mature form of caspase-11 (p20 and p26) by *S. pneumoniae* infection was decreased in NLRP6^−/−^ and ASC^−/−^ macrophages (Figure 3E). Similarly, the induction of the maturation of GSDMD (GSDMD-N) was reduced in NLRP6^−/−^ macrophages, but the induction of pro-GSDMD was increased (Figure 3F,G). Taken together, these results showed that NLRP6 inflammasome can be activated by *S. pneumoniae* and during this process, caspase-1 and caspase-11 can be recruited and cleavage pro-IL-1β into IL-1β. In addition, pro-GSDMD was also cleaved to generate an N-terminal product (GSDMD-N) that formed pores in membranes and triggered pyroptosis. 

### 2.4. NLRP6 Negatively Regulates NF-κB and ERK Signal Pathways in S. pneumoniae-Infected Macrophages

To investigate the role of NLRP6 in regulating NF-κB and ERK signal pathways in *S. pneumoniae*-infected macrophages, the expression of different signaling pathways including NF-κB, IκBα, and ERK1/2 were detected by western blot. The expression of p-p65, p-IκBα was dramatically up-regulated in *S. pneumoniae*-infected macrophages. Similarly, the expression of p-ERK1/2 was also significantly increased after 15 min and 30 min stimulation (Figure 4, * *p* < 0.05). These results suggest that *S. pneumoniae* effectively activates the NF-κB and ERK signaling pathways in macrophages. However, the expression of p-p65 and p-IκBα was significantly higher in *S. pneumoniae*-infected NLRP6^−/−^ macrophages compared with WT macrophages (Figure 4B,D, * *p* < 0.05) while the expression of p-ERK1/2 was only markedly increased in NLRP6^−/−^ macrophages at 15 min and 30 min post infection (Figure 4C). These results suggest that NLRP6 negatively regulate the activation of NF-κB and ERK signaling pathways during macrophages infection with *S. pneumoniae*. 

### 2.5. NLRP6 Damages the Host in S. pneumoniae Infection

To identify the role of NLRP6 in the host defense against *S. pneumoniae* infection in vivo, an intranasal infection of *S. pneumoniae* in WT and NLRP6^−/−^ mice was performed. The survival curve, bacterial numbers in the lungs and H&E staining were used to determine the role of NLRP6 against *S. pneumoniae* infection. The results demonstrated that 76.5% (13/17) of WT mice died in 2 weeks, but NLRP6^−/−^ mice showed a lower mortality at 46.7% (7/15) (Figure 5A). The number of bacterial numbers in the lungs of WT mice were much higher than in those of NLRP6^−/−^ mice at 48 h post infection (Figure 5B). Furthermore, H&E staining showed that the lungs of WT mice had more severe inflammatory response (Figure 5C). These results indicate that NLRP6 plays a negative role in the host defense against *S. pneumoniae*.

### 2.6. NLRP6 Regulates S. pneumoniae-Induced Immune Response In Vivo

To further investigate the mechanism by which NLRP6^−/−^ mice are resistant to *S. pneumoniae* infection, the levels of cytokines in BALF of WT and NLRP6^−/−^ mice were measured at 12 h post infection. Furthermore, the number of neutrophils and macrophages were determined by flow cytometry. The results showed that the secretion levels of IL-1β and IL-6 in BALF of NLRP6^−/−^ mice were significantly lower than that of WT mice (Figure 6A,C), while the level of TNF-α (Figure 6B) showed no difference. The flow cytometric analysis showed that the number of neutrophils (Figure 6D) and macrophages (Figure 6E) were significantly increased in *S. pneumoniae*-infected NLRP6^−/−^ mice. These results indicate that NLRP6 is required for the production of IL-1β and IL-6 in the host response to *S. pneumoniae* infection *in vivo* and NLRP6^−/−^ mice enhance host defense against *S. pneumoniae* may by the recruitment of neutrophils and macrophages.

## 3. Discussion

*S. pneumoniae* is an important pathogen causing lung diseases, leading to high morbidity and mortality in both immunocompetent and immunocompromised individuals. Thus far, treatment of bacterial pneumonia mainly relies on antibiotics, but the increase of antibiotic resistant bacterial strains makes the treatment less effective. Understanding the host innate immune response will contribute to the development of alternative therapeutic approaches. Inflammasomes are complex proteins and have been characterized by a critical role in clearance of invading bacteria in the host. NLRP6 is a newly and specially characterized member of the NLRs family, which prevents the occurrence of diseases such as colorectal tumors and participates in the regulation of intestinal flora [24]. It has been reported that NLRP6 is highly expressed in intestinal epithelial cells where the activation of caspase-1 and ASC is also detected [18]. Furthermore, NLRP6 is also expressed in different immune cells such as macrophages and dendritic cells [20], indicating the key role of NLRP6 in the host. Although the importance of NLRP6 in the host has been studied, the role of NLRP6 in the lung inflammation is less studied. Therefore, in this study, we investigated the role of NLRP6 in the host against *S. pneumoniae*.

Immune cells including macrophages can trigger acute inflammation and secrete IL-1β, which plays an important role in protecting the host from pneumococcal pneumonia [25,26]. At the same time, IL-1β can mediate the expression of chemokines and the synthesis of fibrinogen as well as adhesion molecules to limit the spread of bacteria, resulting in a decreased burden of bacteria [27,28,29]. It has been reported that IL-1β secretion is required the activation of inflammasomes. Hara et al. showed that NLRP6 is involved in the secretion of IL-1β, IL-18 and IL-6 during *L. monocytogenes*- and *S. aureus*-infected bone-marrow-derived macrophages (BMDMs) [21]. Similarly, our study showed that NLRP6 mediates the secretion of IL-1β and IL-6 during macrophages infection with *S. pneumoniae*. However, knockout of NLRP6 did not affect the *S. pneumoniae*-induced expression of IL-1β mRNA and formation of total IL-1β in this study, which is consistent with the study that found blocking NLRP6 did not change LPS-induced expression of IL-1β mRNA and protein [30]. In our previous studies, *S. pneumoniae*-induced secretion of IL-1β also depended on the activation of AIM2 and NLRP3 inflammasomes [12,16,31]. In this study, in spite of knockout of NLRP6 in the macrophages, the existence of other inflammasomes such as NLRP3 and AIM2 might contribute to the unchanged IL-1β during *S. pneumoniae* infection. In contrast with our’s and Hara’s results that knockout of NLRP6 reduced the IL-6 production, Anand et al. found that knockout of NLRP6 increased *L. monocytogenes*-induced IL-6 expression [20]. Although the secretion of IL-6 is inflammasome-independent, it is also mediated by NLRP6 in different microbial challenges. Van Scheppingen J, et al. showed that the decrease of IL-1β may lead to the reduction of IL-6 [32]. Therefore, the decrease of IL-6 might be post-transductional modifications or other possibilities. However, the exact mechanism underlying decreased IL-6 and IL-12p40 in *S. pneumoniae*-infected NLRP6^−/−^ macrophages and mice needs to be further explored.

It has been shown that overexpression of NLRP6 induced activation of caspase-1 and GSDMD leads to secretion of IL-1β and IL-18 [33]. Furthermore, Hara et al. found that lipoteichoic acid (LTA)-induced activation of NLRP6 promoted processing of caspase-11 and the activation of capase-1, resulting in secretion of IL-1β and IL-18. During this process, the adaptor ASC plays an important role in the recruitment of caspase-1 and caspase-11 [21]. Similarly, our study also suggested that NLRP6 inflammasome induced by *S. pneumoniae* can promote the activation of caspase-11, caspase-1, and GSDMD. Notably, knocking out ASC remarkably decreased the activation of caspase-11, suggesting that NLRP6-mediated activation of caspase-11 is dependent on the formation of ASC.

NF-κB is a family of nuclear transcriptional regulators and plays an important role in regulating initial inflammatory response. It has been identified that NLRP6 specifically inhibited the activation of NF-κB and ERK signaling pathways in response to *L. monocytogenes*, Pam3CSK4, or LPS [20]. Moreover, NLRP6 deficiency leads to upregulation of p-p38 MAPK, p-ERK, and p-IκBα in some diseases, such as allogeneic hematopoietic stem cell transplantation (allo-HSCT) [34], peripheral nerve injury [35], and acute kidney injury (AKI) [36]. These results are similar to our study in which the expression of p-p65, p-IκBα, and p-ERK induced by *S. pneumoniae* was downregulated in NLRP6^−/−^ macrophages, suggesting NLRP6 negatively regulates the inflammatory NF-κB and ERK signaling pathways.

Several groups have reported that after *S. aureus* infection the survival of NLRP6^−/−^ mice was significantly higher than WT mice [21,22], which is similar to our results that *S. pneumoniae*-infected NLRP6^−/−^ mice had lower mortality. In our current study, NLRP6^−/−^ mice showed lower bacterial loads and milder inflammation in the lung after *S. pneumoniae* infection compare with WT mice, indicating the negative role of NLRP6 in the clearance of bacteria. Our results confirmed that NLRP6 knockout increased the clearance of *S. pneumoniae* by increasing the recruitment of neutrophils and macrophages. A similar phenomenon was found in NLRP6^−/−^ mice infected with *S. aureus*, which showed that NLRP6 serves as a negative regulator in the host by modulating the recruitment of neutrophils [22]. Importantly, consistent with the results of in vitro experiments, the secretion of IL-1β and IL-6 were also decreased in the BALF of NLRP6^−/−^ mice. It has been reported that NLRP6^−/−^ cells produced elevated levels of NF-κB- and MAPK-dependent cytokines and chemokines against pathogens [21]. In this study, NLRP6-mediated inhibition of inflammatory signaling pathways in vitro probably leads to damage of the inflammatory response after *S. pneumoniae* infection. However, further study will be needed to explore the NLRP6-mediated inhibition of NF-κB and ERK signal pathways in vivo.

In conclusion, we investigated the role of NLRP6 in the host against *S. pneumoniae* both in vivo and in vitro. NLRP6 mediated *S. pneumoniae*-induced secretion of IL-1β, which was dependent on activation of caspase-1 and caspase-11. However, NLRP6 negatively regulated the inflammatory signaling pathway in *S. pneumoniae*-infected macrophages. Furthermore, NLRP6 deficient mice showed low mortality against *S. pneumoniae* infection, indicating the negative role of NLRP6 in regulating inflammatory response against microbial infection. These findings provide further insights into the role of NLRP6 inflammasome-mediated inflammatory response in the host defense against *S. pneumoniae* infection.

## 4. Materials and Methods

### 4.1. Mice

Eight to ten-week-old WT C57BL/6 mice were purchased from the Chongqing Academy of Chinese Materia Medical (Chongqing, China). Equal age- and gender- matched NLRP6^−/−^ mice were kindly provided by Feng Shao from the National Institute of Biological Sciences (Beijing, China). All gene knockout mice were on the C57BL/6 background and were maintained under specific pathogen-free (SPF) conditions. This study was approved by the Institutional Animal Care and Use Committee of Southwest University (IACUC-2019-0923-08).

### 4.2. Bacteria

*Streptococcus pneumoniae* D39 (serotype 2) were kindly gifted by Kohsuke Tsuchiya (Kanazawa University, Japan). *S. pneumoniae* were grown overnight on tryptic soy agar (Hope Biotech, Qingdao, China) with 5% (*v*/*v*) defibrinated sheep blood, then a single colony was inoculated into Todd–Hewitt broth (Hope Biotech, Qingdao, China) supplemented with 0.5% yeast extract (THY). Subsequently, bacteria media were incubated until log-phase (optical density at 600 nm [OD_600_] = 0.5) and centrifuged at 6000 rpm at 4 °C for 20 min. Finally, the bacterial pellet was suspended in phosphate-buffered saline (PBS) with 10% glycerinum and stored at −80 °C. The concentration was determined by counting the colonies on blood agar plates.

### 4.3. In Vivo Infection Experiments

WT and NLRP6^−/−^ mice were anesthetized with pentobarbitone (MREDA, Beijing, China) and then infected intranasally with 5 × 10^7^ CFU bacteria in 20 μL PBS [12]. Lung tissues were collected at 48 h post infection and homogenized in PBS which were serially diluted and plated onto blood agar after a 12 h culture for colony counting. Lung tissues were fixed with 10% formaldehyde and then embedded in paraffin. After that, the paraffin sections were stained by hematoxylin and eosin (H&E) and observed under microscopy. Survival mice was monitored every day until 2 weeks post infection. To reduce the potential for prolonged suffering, the infected mice with irreversible fatal symptoms such as reduced activity, severe coughing, hunched posture, piloerection, and/or tachypnoea [37] were euthanized. BALF samples were collected 12 h post infection according in 1ml PBS. BALF from non-infected WT and NLRP6^−/−^ mice was collected as control.

### 4.4. Flow Cytometric Analysis

After washing the BALF, the total number of cells was counted. Then, cells were stained with PE anti-mouse F4/80 Antibody (BioLegend, San Diego, CA, USA) and FITC anti-mouse Ly-6G/Ly-6C (Gr-1) Antibody (BioLegend, San Diego, CA, USA) in an FACS buffer (0.5% bovine serum albumin [BSA] in PBS) at 4 °C for 30 min. Data were analyzed on the NovoExpress. The number of neutrophils and macrophages were calculated.

### 4.5. Macrophages

Mice were intraperitoneally injected with 2 mL of 4% thioglycolate medium (Eiken Chemical, Tokyo, Japan), and 3 days later peritoneal exudate cells (PECs) were obtained by peritoneal lavage, as reported previously. The mice PECs were washed by PRMI 1640 medium (Gibco, Gaithersburg, MD, USA) and suspended in RPMI 1640 with 10% fetal calf serum (FCS). The cells were seeded at a density of 2 × 10^5^ cells/well in 48-well plates or at a density of 1 × 10^6^ cells/well in 12-well plates at 37 °C plus 5% CO_2_ and incubated for at least 2 h. After incubation, nonadherent cells were removed and the adherent cells were used for in vitro studies. More than 95% of the adherent PECs were F4/80-stained positive cells, as determined by flow cytometry.

### 4.6. ELISA

Macrophages were infected with *S. pneumonia* at a multiplicity of infection (MOI) for 6 h, and then 100 μg/mL gentamicin (Beyotime, Beijing, China) was added and cultured for 18 h. After 24 h incubation, supernatants were collected, and cytokines were determined by ELISA according to the manufacturer’s instructions. ELISA kits used in this study contained TNF-α, IL-6, IL-1β, IL-1α, and IL-12p40 and were purchased from Invitrogen (CA, USA).

### 4.7. Western Blot Analysis

Cells were cultured on 12-well plates at a density of 1 × 10^6^ cells/well in an RPMI 1640 medium supplemented with FCS for at least 2 h. Then, the medium was replaced with Opti-MEM (Invitrogen). The cells were infected with D39 for 6 h at an MOI of 1, and then 100 μg/mL gentamicin was added to continue the culture for 18 h. After 24 h infection, supernatants were collected and cells were lysed with a radio-immunoprecipitation assay (RIPA) buffer (Beyotime, Beijing, China). To determine phosphorylation of NF-κB p65, IκBα, and ERK1/2 the macrophages were infected with the *S. pneumoniae* D39 strain at an MOI of 1 for 0, 15, 30, 60, or 120 min. Then, the macrophages were lysed by the RIPA buffer with PMSF (Beyotime, Beijing, China). The supernatants were concentrated 20 folds with 20% (*w*/*v*) trichloroacetic acid (TCA) and concentrations were determined using a BCA protein detection kit (Beyotime, Beijing, China). The supernatants and the cell lysates were separated by a 10–15% SDS-PAGE gel and subsequently transferred to polyvinylidene difluoride (PVDF) membranes for 2 h. The membranes were immunoblotted with anti-Caspase1-p20 Ab (AdipoGen, AG20B-0042), anti-IL-1β Ab (Bioss, Beijing, China), anti-Caspase11-p20 Ab (Santa cruz biotechnology, Dallas, TX USA), anti-GSDMD Ab (Abcam, Cambridge, UK), anti-ASC Ab (Cell signaling technology, Danvers, MA, USA), anti-NF-κB p65 Ab (Bioss, Beijing, China), anti-phospho-NF-κB p65 Ab (Beyotime, Beijing, China), anti-IκBα Ab (Proteintech, Rosemont, IL, USA), anti-phospho-IκBα Ab (Wanleibio, Shenyang, China), anti-ERK1/2 Ab (Bioss, Beijing, China), anti-phospho-ERK1/2 Ab (Cell signaling technology, Danvers, MA, USA), and anti-GAPDH monoclonal antibody (Beyotime, Beijing, China).

### 4.8. ASC Oligomerization

PECs were cultured in 12-well plates at a density of 1 × 10^6^ cells/well for at least 2 h. After removing nonadherent cells, the adherent PECs were infected with D39 for 6 h, and then gentamicin was added to continue the culture for 18 h. Subsequently, the supernatants were collected, and the cells were lysed with cold PBS containing 0.5% Triton X-100. The cell lysates were centrifuged at 13,000 rpm for 15 min at 4 °C to obtain the cell pellets. The pellets were washed twice with cold PBS and suspended in 200 μL PBS. The resuspended pellets were cross-linked with 2 mM fresh disuccinimidyl suberate (DSS) at 37 °C for 30 min and then the pellets were centrifuged at 13,000 rpm for 15 min at 4 °C. The cross-linked pellets were dissolved in 30 μL 1 × SDS-PAGE sample loading buffer and samples were boiled for 5 min before the western blot analysis.

### 4.9. Quantitative RT-PCR

Macrophages were infected with the *S. pneumoniae* D39 at an MOI of 1 for 9 h. Total RNA was extracted by using the RNA Pre-Pure kit (Tiangen, Beijing, China). Complementary DNA was then synthesized by PrimeScript RT reagent Kit (TaKaRa, Dalian, China) according to the manufacturer’s instructions. Quantitative real-time RT-PCR was carried out using SsoFast Eva Green Super-Mix (Bio-Rad, Hercules, CA, USA) and performed on a Bio-Rad CFX 96 instrument. Primers were used as follows:

*IL-1β* forward 5′- GAA ATG CCA CCT TTT GAC AGT G and reverse 5′- TGG ATG CTC TCA TCA GGA CAG, *β-actin* forward 5′-TGG AAT CCT GTG GCA TCC ATG AAA C, and reverse 5′- TAA AAC GCA GCT CAG TAA CAG TCC G.

### 4.10. Statistical Analysis

Statistical analysis was performed using GraphPad Prism software v6 (San Diego, CA, USA), and the data are presented as mean ± SD. Student’s *t*-test was used to analyze the statistical differences for comparisons between two groups. Statistical analysis for survival curves was performed using the log-rank test. Statistical significance was determined as *p* < 0.05.

## Figures and Tables

**Figure 1 ijms-22-03876-f001:**
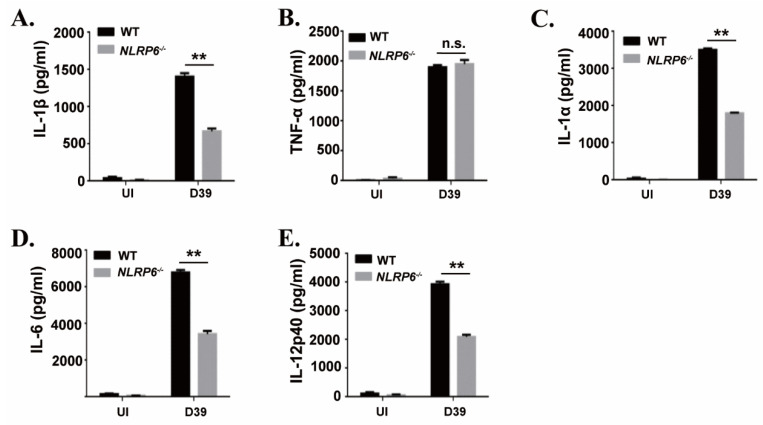
NLR family pyrin domain containing 6 (NLRP6) mediates inflammatory cytokines expression in *S. pneumoniae*-infected macrophages. Macrophages from WT or NLRP6^−/−^ mice were uninfected or infected with *S. pneumoniae* at a multiplicity of infection (MOI) of 1 for 6 h. Then, gentamicin (100 μg/mL) was added to the cultures and incubated for 18 h. After 24 h infection, the supernatants were collected and the levels of IL-1β (**A**), TNF-α (**B**), IL-1α (**C**), IL-6 (**D**) and IL-12p40 (**E**) in the supernatants were determined by ELISA. All of the experiments were independently performed three times. Statistical significance was determined by Student’s *t*-test (** *p* < 0.01).

**Figure 2 ijms-22-03876-f002:**
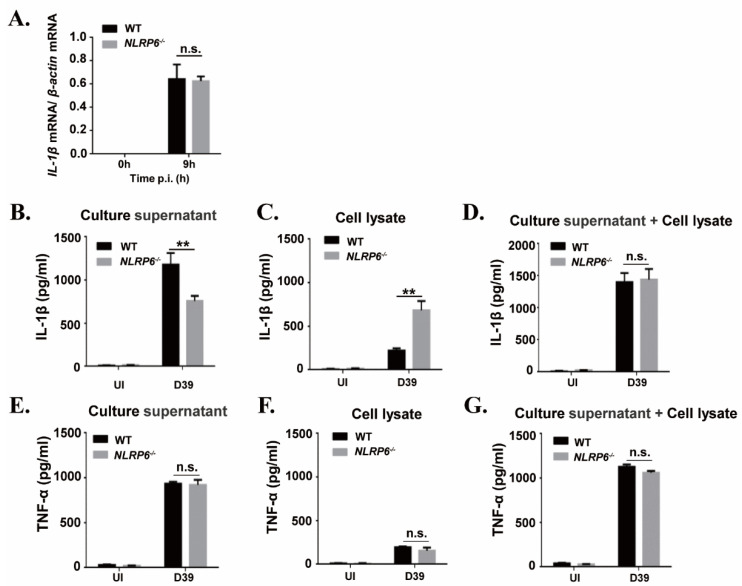
NLRP6 mediates maturation and secretion of IL-1β but does not regulate the induction of IL-1β mRNA in *S. pneumoniae*-infected macrophages. Macrophages were infected with the *S. pneumoniae* at an MOI of 1. (**A**) Total cellular RNA was extracted 9 h after infection, and the level of IL-1β mRNA expression was analyzed by real-time RT-PCR. The levels of IL-1β (**B**–**D**) and TNF-α (**E**–**G**) in the culture supernatants and cell lysates were determined by ELISA at 24 h post infection. All of the experiments independently performed three times. Statistical significance was determined by Student’s *t*-test (** *p* < 0.01).

**Figure 3 ijms-22-03876-f003:**
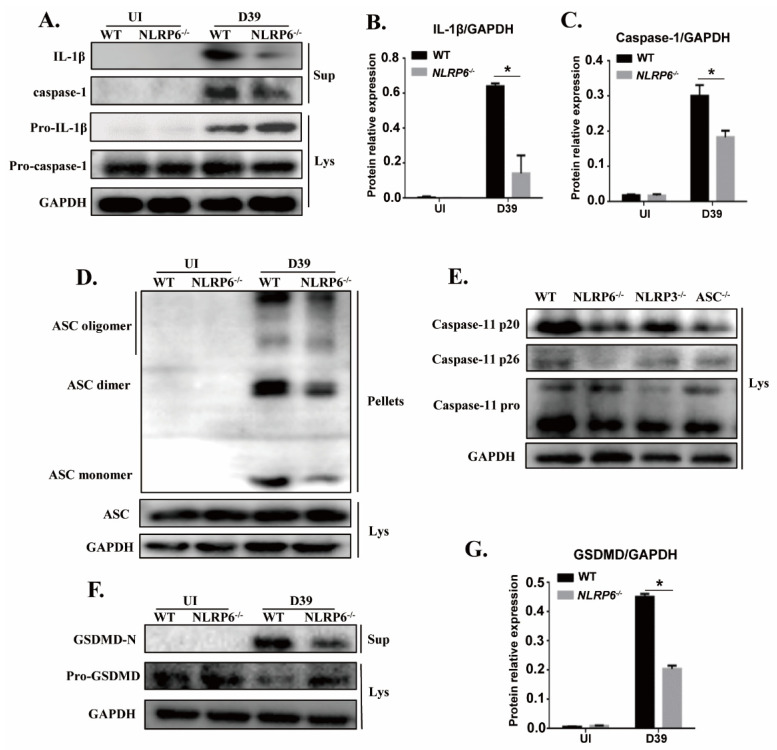
NLRP6 inflammasome mediates activation of caspase-1 and caspase-11 in *S. pneumoniae*-infected macrophages. The culture supernatant (Sup) and the cell lysate (Lys) in *S. pneumoniae*-infected macrophages of WT and NLRP6^−/−^ mice were collected at 24 h post infection. Western blotting analysis was used to detect the formation of caspase-1 (p20: subunit; p45: precursor) (**A**), caspase-11 (**E**), and GSDMD (**F**). Immunoblotting was used to analyze the formation of ASC (**D**). Ratio of IL-1β (**B**), caspase-1 (**C**), and GSDMD-N (**G**) levels against GAPDH levels was quantified. Statistical significance was determined by Student’s *t*-test. (* *p* < 0.05).

**Figure 4 ijms-22-03876-f004:**
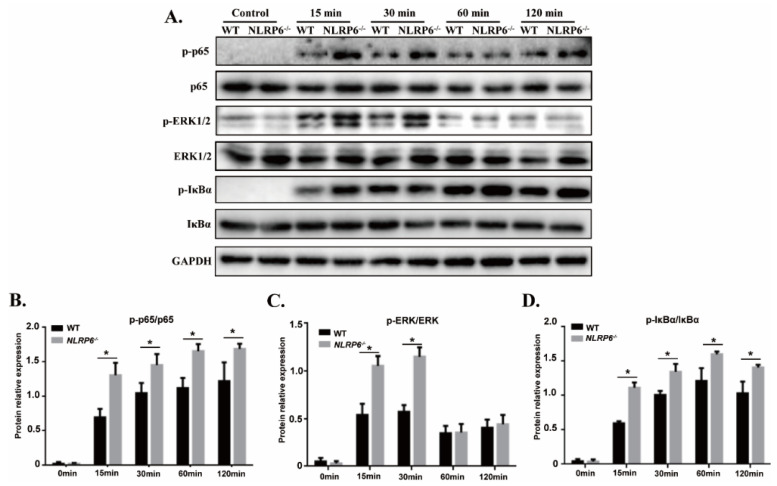
NLRP6 inhibits activation of NF-κB and ERK signal pathways in *S. pneumoniae*-infected macrophages. Macrophages from WT and NLRP6^−/−^ were infected with *S. pneumoniae* at an MOI of 1 for 0, 15, 30, 60, and 120 min. (**A**) p-p65, pERK, pIκBα, p65, ERK, and IκBα protein expression was determined by western blot. Ratio of p-p65 (**B**), pERK (**C**), and pIκBα (**D**) levels against p65, ERK, and IκBα levels was quantified. Statistical significance was determined by Student’s *t*-test. (* *p* < 0.05).

**Figure 5 ijms-22-03876-f005:**
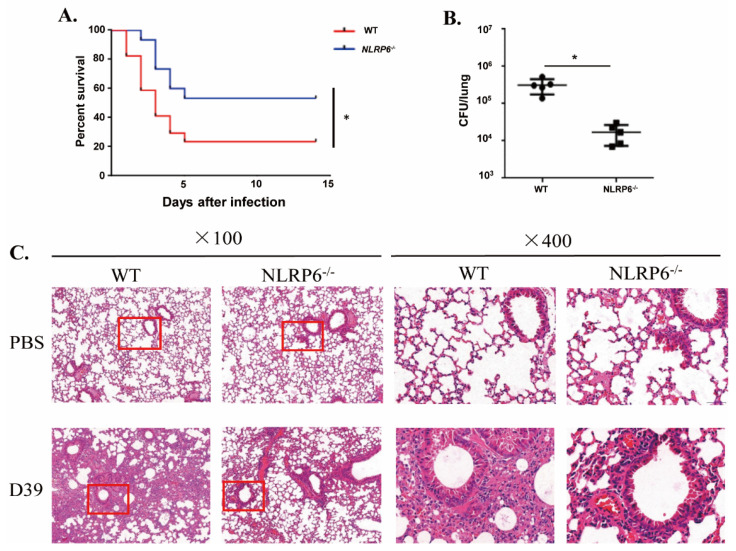
NLRP6^−/−^ mice are resistant to *S. pneumoniae* infection. WT and NLRP6^−/−^ mice were intranasally infected with *S. pneumoniae* (5 × 10^7^ CFU). (**A**) WT (*n* = 17) and NLRP6^−/−^ (*n* = 15) mice were monitored for 14 d. (**B**) Lungs were homogenized and colony counting was performed at 48 h post infection in *S. pneumoniae*-infected WT (*n* = 5) and NLRP6^−/−^ (*n* = 5) mice. (**C**) The lungs were collected at 48 h post infection for hematoxylin and eosin staining (original magnification ×100 and ×400). Pictures are representative of three mice from each group of mice. Tests for statistical significance were performed using one-way ANOVA followed by the Bonferroni test (* *p* < 0.05).

**Figure 6 ijms-22-03876-f006:**
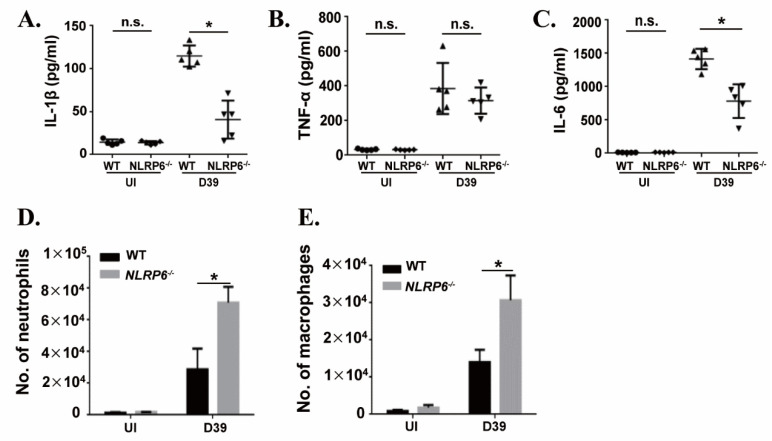
NLRP6 regulates *S. pneumoniae*-induced immune response in vivo. WT and NLRP6^−/−^ mice were uninfected or intranasally challenged with *S. pneumoniae* (5 × 10^7^ CFU), and bronchoalveolar lavage fluid (BALF) was collected at 12 h post infection. The levels of IL-1β (**A**), TNF-α (**B**), and IL-6 (**C**) were measured by ELISA (*n* = 5). Quantification of the number of neutrophils (**D**) and the number of macrophages (**E**) (*n* = 2–5) were determined by flow cytometry. Tests for statistical significance were performed using one-way ANOVA followed by the Bonferroni test (* *p* < 0.05).

## Data Availability

All the data generated for this study are included in the article.

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
