# Peer review of "The Critical Role of NLRP6 Inflammasome in Streptococcus pneumoniae Infection In Vitro and In Vivo"

_ijms, 2021, doi:10.3390/ijms22083876_

Round 1

Reviewer 1 Report

The study performed by Xu and Wu et al. shows the negative role of the NLRP6 inflammasome after Streptococcus pneumonia infection. To conclude this, authors use both in vitro and in vivo experiments in an elegant way.

As the authors mention in the introduction, the role of NLRP6 against gram-positive bacteria involved in lung diseases has been published before (https://doi.org/10.1371/journal. ppat.1007308) using S. aureus as pathogen with the same results. This may detract from the novelty of the study, but supports the results obtained by Ghimire et al.

In this regard I have some minor revisions or comments:

Line 51-52: When the authors say that NLRs have been recognized for the critical role in innate immunity against microbial infection, they should add some references to support this.

Line 78: This study is not the first about the NLRP6 in the host against pulmonary microbial infection. Ref. 18 of this study supports this.

In the introduction: I suggest add a little literature about the canonical and non-canonical inflammasome activation due to the authors work with caspase-1 and caspase-11.

Fig 1: I miss some LDH data accompanying the ELISA assays in macrophages. Maybe these data could be presented in supplementary data.

Fig 2: Could the authors show the mRNA levels for IL-6 and TNFa? If the mRNA expression is not affected, maybe they could discus if the less amount of IL-6 protein release could be due to post transductional modifications or other possibilities. Why is less amount of IL-6 if NF-kB is increased in NLRP6-/- macrophages and as far as I know the release of IL-6 is inflammasome- independent.

Line 230: References supporting this statement could be added: “Caspase-11 is also critical for the activation….”

Line 237-Linked with Fig 3A: the authors could quantify the protein levels against GAPDH as they make in figure 4 to make this clearer. In this regard the authors could perform an enzymatic assay to measure the caspase-1 activity to support the results they authors mention.

Fig 3B: The authors could perform an immunocytochemistry against ASC to support the data showed.

In the discussion the authors contemplate the possible role of others inflammasomes (NLRP4 or NLRC4) as possible NLRP6 partners in this specific infection. This is a very interesting point that maybe could be more detailed. The authors maybe could hypostatize why in NLRP6-/- mice the amount of pathogens is reduced compared with WT mice. Could be a role for IFNg?

Author Response

The study performed by Xu and Wu et al. shows the negative role of the NLRP6
inflammasome after Streptococcus pneumonia infection. To conclude this, authors use
both in vitro and in vivo experiments in an elegant way.
As the authors mention in the introduction, the role of NLRP6 against gram-positive
bacteria involved in lung diseases has been published before
(https://doi.org/10.1371/journal. ppat.1007308) using S. aureus as pathogen with the
same results. This may detract from the novelty of the study, but supports the results
obtained by Ghimire et al.
In this regard I have some minor revisions or comments:
Point 1: Line 51-52: When the authors say that NLRs have been recognized for the
critical role in innate immunity against microbial infection, they should add some
references to support this.
Response 1: Thanks for your comments. We already add 2 references (8, 9) to support
it. (line 52)
Point 2: Line 78: This study is not the first about the NLRP6 in the host against
pulmonary microbial infection. Ref. 18 of this study supports this.
Response 2: Thanks for your comments. We already changed the words to indicate that
our study is not the first study about the role of NLRP6 in the host against pulmonary
microbial infection. Indeed, our study provide more information and confirm the role
of NLRP6 in the pulmonary infection. It will be also useful for the further study. (line
86)
Point 3: In the introduction: I suggest add a little literature about the canonical and noncanonical
inflammasome activation due to the authors work with caspase-1 and
caspase-11.
Response 3: Thanks for your comments. We already added some literatures about
canonical and non-canonical inflammasome activation, and also introduce the
difference between canonical and non-canonical inflammasome activation in the
introduction in line 56-63.
Point 4: Fig 1: I miss some LDH data accompanying the ELISA assays in macrophages.
Maybe these data could be presented in supplementary data.
Response 4: Thanks for your comments. I agree with you that it would be better to show some LDH data in supplementary data. However, our data already showed that S. pneumoniae infection induced the activation of caspase-11 and cleavage of GSDMD as shown in figure 3, indicating that S. pneumoniae induced pyroptosis at 24h post infection. Therefore, we did not measure the LDH after infection.
Point 5: Fig 2: Could the authors show the mRNA levels for IL-6 and TNFa? If the mRNA expression is not affected, maybe they could discus if the less amount of IL-6 protein release could be due to post transductional modifications or other possibilities. Why is less amount of IL-6 if NF-kB is increased in NLRP6-/- macrophages and as far as I know the release of IL-6 is inflammasome- independent.
Response 5: Thanks for your comments. We also found this phenomenon. It is very interesting that Hara et al. showed IL-6 also reduced in the S. aureus-infected BMDM from NLRP6-/- mice. Since the secretion of IL-6 is inflammasome-independent, how IL-6 secretion is up or down influenced by NLRP6 deficient in different microbial challenges, we agree it might be post transductional modifications or other possibilities, need to be further explored. It is very valuable to explore around this phenomenon. We already added some discussion on this (line 362 and 369-374). However, our study focused on NLRP6-dependent secretion of IL-1β instead of IL-6.
Point 6: Line 230: References supporting this statement could be added: “Caspase-11 is also critical for the activation….”
Response 6: Thanks for your comments. We already add 2 references (21) to support it (line 239).
Point 7: Line 237-Linked with Fig 3A: the authors could quantify the protein levels against GAPDH as they make in figure 4 to make this clearer. In this regard the authors could perform an enzymatic assay to measure the caspase-1 activity to support the results they authors mention.
Response 7: Thanks for your comments. To make the figure 3 more clearer, we added not only ratio of caspase-1 level against GAPDH level, but also the ratio of IL-1β, GSDMD levels against GAPDH levels. I agree with you that it would be better to measure the caspase-1 activity to quantify them better. In our study, the western-blot result directly showed the activation of caspase-1, so we did not use enzyme assay to measure caspase-1 activity.
Point 8: Fig 3B: The authors could perform an immunocytochemistry against ASC to support the data showed.
Response 8: Thanks for your comments. Performing an immunocytochemistry against ASC would be better, but there was a significantly difference about ASC monomer, dimer and oligomer between WT and NLRP6-/- macrophages, which already indicates that NLRP6 has an effect on the ASC.
Point 9: In the discussion the authors contemplate the possible role of others inflammasomes (NLRP4 or NLRC4) as possible NLRP6 partners in this specific infection. This is a very interesting point that maybe could be more detailed. The authors maybe could hypostatize why in NLRP6-/- mice the amount of pathogens is reduced compared with WT mice. Could be a role for IFNg?
Response 9: Thanks for your comments. It is a nice point to you mentioned the possibilities of NLRP6 partners in this infection. It has been reported that NLRC4 inflammasome cannot be activated by S. pneumoniae-infection. Actually, there is very few literatures to show combination of inflammasomes. I agree with you that IFN-γ could play a role to protect the NLRP6-/- mice. Because Ghimire et al. reported that NLRP6 KO mice displayed increased neutrophil recruitment following infection of S. aureus, and neutrophils from the KO mice demonstrated enhanced intracellular bacterial killing, increased ROS and IFN-γ production. That is a good point to discuss it. We already add some possibilities of reduced number of bacteria in the discussion section in line 385-388.

Reviewer 2 Report

Xu, D. et al. The critical role of NLRP6 inflammasome in Streptococcus pneumoniae infection in vitro an in vivo.

This manuscript explores the role of NLRP6 inflammasome in the inflammatory response triggered by Streptococcus pneumoniae (S. pneumoniae) infection. Using NLRP6 deficient mice and isolation of peritoneal macrophages the authors found that in the absence of NLRP6 inflammasome macrophages fail to secrete IL-1β, IL-1α, IL-6 and IL-12, while the secretion of TNF-α remains the same. Further description of the mechanism involved in the lack of IL-1β secretion showed that the absence of NLRP6 reduces the activation of caspase 11 and somewhat reduces the activation of caspase 1. In addition, conversion of pro-gasdermin into gasdermin-N was also reduced in S. pneumoniae infected NLRP6-/- macrophages. NLRP6 was also important to downregulate the levels of p-p65, p-IkBα, and p-ERK in S. pneumoniae infected macrophages. Finally, the increased survival of NLRP6-deficient mice to intranasal infection with S. pneumoniae suggests that the formation and activation of the NLRP6 inflammasome has a detrimental effect on the host.

This manuscript shows interesting data regarding the mechanisms through which NLRP6 might contribute to the host immune response against S. pneumoniae and supports the idea of NLRP6 inflammasomes mediate lung damage detrimental to the host. However, the data presented confirms previously published data where NLRP6 inflammasome increases local inflammation and morbidity to pulmonary infections with gram-positive bacteria, thus lacks novelty.

Since NLRP function seems to fine tune local innate immune response and their effects are context dependent, the current manuscript would benefit from experiments that analyze the role of NLRP6 in alveolar and lung parenchyma macrophages in vitro, and the local cytokine production in the lungs (both in naïve and infected mice). An interesting finding was the reduced secretion of IL-6 and IL-12, coupled with the changes in NF-KB activation, however experiments aimed to determine whether these data are related or if it is secondary to reduced IL-1b secretion were not attempted. Also, it would be helpful to add more information on the type of IL-12 detected (p40, p70, or p35).

Author Response

Comments and Suggestions for Authors 2
This manuscript explores the role of NLRP6 inflammasome in the inflammatory response triggered by Streptococcus pneumoniae (S. pneumoniae) infection. Using NLRP6 deficient mice and isolation of peritoneal macrophages the authors found that in the absence of NLRP6 inflammasome macrophages fail to secrete IL-1β, IL-1α, IL-6 and IL-12, while the secretion of TNF-α remains the same. Further description of the mechanism involved in the lack of IL-1β secretion showed that the absence of NLRP6 reduces the activation of caspase 11 and somewhat reduces the activation of caspase 1. In addition, conversion of pro-gasdermin into gasdermin-N was also reduced in S. pneumoniae infected NLRP6-/- macrophages. NLRP6 was also important to downregulate the levels of p-p65, p-IkBα, and p-ERK in S. pneumoniae infected macrophages. Finally, the increased survival of NLRP6-deficient mice to intranasal infection with S. pneumoniae suggests that the formation and activation of the NLRP6 inflammasome has a detrimental effect on the host.
This manuscript shows interesting data regarding the mechanisms through which NLRP6 might contribute to the host immune response against S. pneumoniae and supports the idea of NLRP6 inflammasomes mediate lung damage detrimental to the host. However, the data presented confirms previously published data where NLRP6 inflammasome increases local inflammation and morbidity to pulmonary infections with gram-positive bacteria, thus lacks novelty.
Since NLRP function seems to fine tune local innate immune response and their effects are context dependent, the current manuscript would benefit from experiments that analyze the role of NLRP6 in alveolar and lung parenchyma macrophages in vitro, and the local cytokine production in the lungs (both in naïve and infected mice). An interesting finding was the reduced secretion of IL-6 and IL-12, coupled with the changes in NF-KB activation, however experiments aimed to determine whether these data are related or if it is secondary to reduced IL-1b secretion were not attempted. Also, it would be helpful to add more information on the type of IL-12 detected (p40, p70, or p35).

Response 1: Thanks for your comments. We already added the type of IL-12p40 and changed the figure1 E. Our study aimed to determine the role of NLRP6 in the host against S. pneumoniae infection. Secretion of IL-1β is dependent on the activation of NLRP6 inflammasome. We also found the phenomenon that secretion of IL-6 and IL-
12p40 were reduced in S. pneumoniae-infected PECs from NLRP6-/- mice. As far as we
known, secretion of IL-6 and IL-12 is not dependent on inflammasome activation. It is
very interesting that Hara et al. showed IL-6, IL-12p40 also reduced in the S. aureusinfected
BMDM from NLRP6-/- mice. How IL-6 and IL-12p40 secretion are up or down
influenced by NLRP6 deficient in different microbial challenges, we guess it might be post transductional modifications or other possibilities, need to be further explored. It is very valuable to explore around this phenomenon. We already added some discussion on this (line 362 and 369-374). However, our study focused on NLRP6-dependent secretion of IL-1β instead of IL-6 and IL-12p40.

Reviewer 3 Report

The manuscript by Xu et al investigates the roles of NLRP6 during S. pneumoniae-induced pneumonia. Although the result is similar to one paper from Jeyaseelan Lab (reference 22 in the manuscript) which also shows negative role of NLRP6 in the pathogenesis of Gram-positive bacteria (S. aureus) induced pneumonia, this is the  first study on investigating the roles of NLRP6 inflammasome during S. pneumoniae-induced pneumonia. The strong point is the use of both in vitro and in vivo techniques.  The weak point is the lack of mechanisms to explain the in vivo results.

Major comments:

  1. The authors have shown that NLRP6 KO mice had better survival compared to that of WT mice along with reduction in bacterial burden and attenuated inflammation in the lungs. However, what caused higher bacterial clearance in the lungs is not shown. Was the neutrophils/macrophages recruitment high? or did the KO mice had less pyroptosis compared to WT mice? The authors have described regulation of NFkB pathways by NLRP6 as a contributor to the host defense in the discussion section. However, the suppression of such pathways in the lungs is not shown in the paper.
  2. The cytokines production in the lungs (or broncho alveolar lavage fluid) specially IL-1B, TNF-alpha, and IL-6 should also be measured to confirm less inflammation in the lungs.

Minor comments:

None

Author Response

Major comments: The authors have shown that NLRP6 KO mice had better survival compared to that of WT mice along with reduction in bacterial burden and attenuated inflammation in the lungs. However, what caused higher bacterial clearance in the lungs is not shown. Was the neutrophils/macrophages recruitment high? or did the KO mice had less pyroptosis compared to WT mice? The authors have described regulation of NF-kB pathways by NLRP6 as a contributor to the host defense in the discussion section. However, the suppression of such pathways in the lungs is not shown in the paper. Response: Thanks for reviewer’s comments and suggestions. Recently, we carried out the in vivo experiments accordingly in manuscript (Figure 6) to explain why NLRP6 KO mice had better bacterial clearance and improved survival. We examined the recruitment of neutrophils/macrophages after S. pneumoniae infection which showed that the recruitment of neutrophils/macrophages into the lung in NLRP6-/- mice were significantly higher than that of WT mice after S. pneumoniae infection. We added a paragraph to the results part and some sentences to the abstract, introduction and discussion, as follows: “Furthermore, increased bacterial clearance in NLRP6 deficient mice was modulated by the recruitment of macrophages and neutrophils.” Line 30-31
“Moreover, NLRP6 negatively modulated the recruitment of neutrophils and macro-phages.” Line 86-87
“NLRP6 regulates S. pneumoniae-induced immune response in vivo. To further investigate the mechanism by which NLRP6-/- mice are resistant to S. pneumoniae infection, the levels of cytokines in BALF of WT and NLRP6-/- mice were measured at 12 h post infection. Furthermore, the number of neutrophils and macrophages were determined by flow cytometry. The results showed that the secretion levels of IL-1β and IL-6 in BALF of NLRP6-/- mice were significantly lower than that of WT mice (Figure 6A and 6C), while the level of TNF-α (Figure 6B) showed no difference. The flow cytometric analysis showed that the number of neutrophils (Figure 6D) and macrophages (Figure 6E) were significantly increased in S. pneumoniae- infected NLRP6-/- mice. These results indicate that NLRP6 is required for the production of IL-1β and IL-6 in the host response to S. pneumoniae infection in vivo and NLRP6-/- mice enhance host defense against S. pneumoniae may by the recruitment of neutrophils and Line 321-332
“Our results confirmed that NLRP6 knockout increased the clearance of S. pneumoniae by increasing the recruitment of neutrophils and macrophages. Similar phenomenon was found in NLRP6-/- mice infected with S. aureus, which showed that NLRP6 serves as a negative regulator in the host by modulating the recruitment of neutrophils [22].”
Line 401-404
Indeed, we did not show NF-kB pathways in the lung. I agree with you that it would be better to measure it in the lungs. However, according to Figure 6, NLRP6 KO increased the number of neutrophils and macrophages in the lung. Furthermore, the results in figure 4 showed that NLRP6 negatively regulated NF-kB pathways in macrophages, which may indicate the negative role of NLRP6 on regulating NF-kB pathways. Anand et al. found that NLRP6 inhibit the activation of NF-kB pathways in lung upon L. monocytogenes infection (Nature, 2012, 488(7411): 389-393). Whether NLRP6 inhibit the activation of NF-kB pathways in lung upon S. pneumoniae infection needs to be further investigated. So we have already changed our sentences about this discussion as follows: “In this study, NLRP6-mediated inhibition of inflammatory signaling pathway in vitro probably leads to the damage of inflammatory response after S. pneumoniae infection. However, further study will be needed to explore the NLRP6-mediated inhibition of NF-κB and ERK signal pathways in vivo.” Line 409-412
The cytokines production in the lungs (or broncho alveolar lavage fluid) specially IL-1B, TNF-alpha, and IL-6 should also be measured to confirm less inflammation in the lungs. Response: Thanks for your comments. We already measured the production of IL-1β, TNF-α and IL-6 from BALF. The results showed that comparing with WT mice, the secretion of IL-6 and IL-1β in the BALF from NLRP6-/- mice were decreased, while TNF-α was not affected. These are consistent with the data in vitro. We have already added the above mentioned in vivo data in the current manuscript in figure 6, and we added a paragraph to the results part and some sentences to the discussion as follows: “NLRP6 regulates S. pneumoniae-induced immune response in vivo. To further investigate the mechanism by which NLRP6-/- mice are resistant to S. pneumoniae infection, the levels of cytokines in BALF of WT and NLRP6-/- mice were measured at 12 h post infection. Furthermore, the number of neutrophils and macrophages were determined by flow cytometry. The results showed that the secretion levels of IL-1β and IL-6 in BALF of NLRP6-/- mice were significantly lower than that of WT mice (Figure 6A and 6C), while the level of TNF-α (Figure 6B) showed no difference. The flow cytometric analysis showed that the number of neutrophils (Figure 6D) and macrophages (Figure 6E) were significantly increased in S. pneumoniae-infected NLRP6-/- mice. These results indicate that NLRP6 is required for the production of IL-1β and IL-6 in the host response to S. pneumoniae infection in vivo and NLRP6-/- mice enhance host defense against S. pneumoniae may by the recruitment of neutrophils and Line 321-332
“Importantly, in consistent with the results of in vitro experiments, the secretion of IL-1β and IL-6 were also decreased in BALF of NLRP6-/- mice.” Line 404-406
Minor comments: None

Round 2

Reviewer 2 Report

I understand that the main objective of the data presented is to assess the role of NLRP6 inflammasome in macrophage response to S. pneumoniae infection and in particular with secretion of IL-1β, however this data lacks novelty as the role of NLRP inflammasomes, including NLRP6 inflammasome, in mediating IL-1β maturation is well described. It would be more informative if the in vitro data could be linked with the in vivo data either by measuring inflammatory cytokines in the lungs of NLRP6-/- mice or having in vitro data from lung macrophages. If the reduced production of IL-12p40 and IL-6 is secondary to reduced IL-1β would also provide novelty to the manuscript. In its current form it is hard to argue that it provides information that was not expected from previously published data.

Author Response

I understand that the main objective of the data presented is to assess the role of NLRP6 inflammasome in macrophage response to S. pneumoniae infection and in particular with secretion of IL-1β, however this data lacks novelty as the role of NLRP inflammasomes, including NLRP6 inflammasome, in mediating IL-1β maturation is well described. It would be more informative if the in vitro data could be linked with the in vivo data either by measuring inflammatory cytokines in the lungs of NLRP6-/- mice or having in vitro data from lung macrophages. If the reduced production of IL-12p40 and IL-6 is secondary to reduced IL-1β would also provide novelty to the manuscript. In its current form it is hard to argue that it provides information that was not expected from previously published data.
Response: Thanks for the reviewer’s comments and suggestions. We are sorry to say that in our previous revision we did not add in vivo data according to the reviewer’s suggestion. Recently, we carried out the in vivo experiments accordingly, we measured cytokines from BALF, and we examined the recruitment of neutrophils/macrophages after S. pneumoniae infection. The results showed that comparing with WT mice, the secretion of IL-6 and IL-1β in the BALF from NLRP6-/- mice were decreased, while TNF-α was not affected. These are consistent with the data in vitro. Furthermore, the recruitment of neutrophils/macrophages into the lung in NLRP6-/- mice were significantly higher than that of WT mice after S. pneumoniae infection, this could explain the increased clearance of S. pneumoniae in the lung and improved survival of NLRP6-/- mice. We have already added the above mentioned in vivo data in the current manuscript in figure 6, and we added a paragraph to the results part as follows:
“NLRP6 regulates S. pneumoniae-induced immune response in vivo.
To further investigate the mechanism by which NLRP6-/- mice are resistant to S. pneumoniae infection, the levels of cytokines in BALF of WT and NLRP6-/- mice were measured at 12 h post infection. Furthermore, the number of neutrophils and macrophages were determined by flow cytometry. The results showed that the secretion levels of IL-1β and IL-6 in BALF of NLRP6-/- mice were significantly lower than that of WT mice (Figure 6A and 6C), while the level of TNF-α (Figure 6B) showed no difference. The flow cytometric analysis showed that the number of neutrophils (Figure 6D) and macrophages (Figure 6E) were significantly increased in S. pneumoniae-infected NLRP6-/- mice. These results indicate that NLRP6 is required for the production of IL-1β and IL-6 in the host response to S. pneumoniae infection in vivo and NLRP6-/- mice enhance host defense against S. pneumoniae may by the recruitment of neutrophils and Line 321-332
We agree with the reviewer that it will provide novelty to our manuscript if we could explain the reduced IL-6 and IL-12p40 in NLRP6-/-macrophage upon S. pneumoniae infection, which might be secondary to reduced IL-1β. Similar results have already reported by Hara et al. in NLRP6-/- macrophage upon S. aureus and L. monocytogenes infection (Cell, 2018, 175(6): 1651-1664 e1614.). Van Scheppingen J, et al. have previously reported that the decrease of IL-1β may leads to the reduction of IL-6 in human astrocytes (Neural Injury and Functional Reconstruction, 2018, 66(5): 1082-1097).
The secretion of IL-12p40 is associated with the immune response of T cells and it might be related to adaptive immune system. Inflammatory cytokine secretion are very complicated modulated. Whether the reduced IL-6 and IL-12p40 in NLRP6-/-macrophage are secondary to reduced IL-1β upon diverse bacterial infection needs to be further investigated separately. We have added the following sentences to the discussion.
“Although the secretion of IL-6 is inflammasome-independent, it is also mediated by NLRP6 in different microbial challenges. Van Scheppingen J, et al. showed that the decrease of IL-1β may lead to the reduction of IL-6 [33]. So the decrease of IL-6 might be post transductional modifications, or other possibilities. However, the exact mechanism underlying decreased IL-6 and IL-12p40 in S. pneumoniae-infected NLRP6-/- macrophages and mice need to be further explored.” Line 372-377

Reviewer 3 Report

All of my concerns are addressed. The paper looks good now.

Round 3

Reviewer 2 Report

Thank you for providing feedback on my previous comments and new data addressing a possible link between the in vitro data obtained in peritoneal cavity macrophages and the lung environment during S. pneumoniae infection. The differences between WT and NLRP6-/- are striking since the absence of NLPR6-/- greatly increased the recruitment of neutrophils and macrophages to the alveolar space, and presumably help eliminate S. pneumoniae. This data brings to question how NLRP6-/- modifies macrophage sensitivity to chemokines (no changes to the manuscript required).

On the role of IL-12p40, since it can be produced by macrophages it is not always related to the initiation of adaptive immune responses, although that is the most frequently studied function. Also, IL-12p40 is the required subunit for the secretion of IL-12p70, IL-23, and can be secreted as IL-12p40 homodimers. Secretion of IL-12p40 has been associated with an anti-inflammatory function and with autocrine regulation of macrophages and dendritic cells.

Finally, please describe in the methods section which equipment was used for acquisition of flow cytometry data from BALF cell suspensions.